# Synthesis and Spectral Study of a New Family of 2,5-Diaryltriazoles Having Restricted Rotation of the 5-Aryl Substituent

**DOI:** 10.3390/molecules25030480

**Published:** 2020-01-23

**Authors:** Biligma Tsyrenova, Valentine Nenajdenko

**Affiliations:** Department of Chemistry, Lomonosov Moscow State University, 119899 Moscow, Russia; tbiligma@gmail.com

**Keywords:** triazole, azide, synthesis, cyclization, fluorescence, stokes shift, quantum yield, solvatochromism

## Abstract

Efficient synthesis of 2,5-diaryl substituted 4-azido-1,2,3-triazoles by the reaction of sodium azide with dichlorosubstituted diazadienes was demonstrated. The optical properties of the prepared azidotriazoles were studied to reveal a luminescence maximum in the 360–420 nm region. To improve the luminescence quantum yields a family of 4-azido-1,2,3-triazoles bearing *ortho*-propargyloxy substituents in the 5 position was prepared. Subsequent intramolecular thermal cyclization permits to construct additional triazole fragment and obtain unique benzoxazocine derivatives condensed with two triazole rings. This new family of condensed heterocycles has a flattened heterocyclic system structure to provide more conjugation of the 5-aryl fragment with the triazole core. As a result, a new type of UV/“blue light-emitting” materials with better photophysical properties was obtained.

## 1. Introduction

1,2,3-Triazoles are an important class of heterocyclic compounds of both theoretical and practical interest [1,2,3,4]. This type of heterocycles has been known for more than 150 years, however, during the few last decades, 1,2,3-triazoles have become an attractive object of research in heterocyclic chemistry due to their highly efficient synthesis through copper(I)-catalyzed acetylene–azide cycloaddition [5,6,7,8,9,10,11,12,13]. A large number of heterocycles containing an embedded triazole core have attracted considerable attention because of their broad range of pronounced biological activity. For example, 1,2,3-triazole derivatives exhibit anti-inflammatory, anesthetic, antimicrobial, anti-arrhythmic, antitumor, and antiviral properties [14,15,16,17]. In addition, 1,2,3-triazole derivatives are used in agrochemistry as insecticides, fungicides and plant growth regulators [18]. 

Nowadays, there is also significant interest in the development of novel selective approaches to 2-substituted 1,2,3-triazoles. These compounds are particularly attractive since they have excellent fluorophore properties [19,20,21,22,23,24,25]. A great number of fluorophores has been developed over the years, but effective UV/blue-light-emitting molecules are still rare due to their relatively high energy gap, which may cause poor photostability or low quantum efficiency. However, the booming research area, such as OLED display study, in photoactive compounds has generated increasing needs for effective UV/blue-light-emitting molecules. Recently Yan and co-workers [22] reported N-2-aryl-1,2,3-triazoles (Figure 1) as new UV/blue-light-emitting compounds with tunable emission and adjustable Stokes shift through planar intramolecular charge transfer.

1,1-Dichlorodiazadienes are a valuable class of electrophiles. These compounds can be prepared using the reaction of carbon tetrachloride witH-N-substituted hydrazones of aldehydes in the presence of CuCl as a catalyst. [26,27] Recently we demonstrated that these compounds are also interesting diazodyes [28,29]. The reaction of 4,4-dichloro-1,2-diazabuta-1,3-dienes with sodium azide has been found to open straightforward access to extremely rare 1,1-bis-azides (Figure 2). These highly unstable compounds are prone to eliminate a N_2_ molecule to cyclize into 4-azido-1,2,3-triazoles **1** bearing two aryl (heteroaryl) groups at positions 2 and 5. The reaction was found to be very general for the highly efficient synthesis of various 4-azidotriazoles. It was demonstrated that these heterocycles are highly attractive building blocks for subsequent preparation of 1,2,3-triazole- derived compounds [30].

The prepared azidotriazoles **1** [30] contain the same structural pattern as the diaryltriazoles used as UV/blue-light-emitting compounds [22] (Figure 1). Therefore, we decided to study optical properties of 4-azido-1,2,3-triazoles **1**. This study is also devoted to investigation of the synthesis of 4-azido-2,5-diaryl-1,2,3-azidotriazoles prepared from arylhydrazines and *ortho*-propargyloxy-benzaldehydes. The presence in the structure of these compounds of a triple bond and the azido group opens up the possibility of intramolecular cyclization of these compounds to obtain unique condensed heterocycles containing two triazole rings (Figure 3b). Moreover, such structural modification can significantly affect the photophysical properties of the intramolecular cyclization products. In this case free rotation of the aryl fragment in the 5 position is impossible. The main aim of this work was studying of optical properties of 2-aryl 1,2,3-trizoles **1** and synthesis of their rigidified analogues obtained via intramolecular acetylene–azide cyclization (Figure 3b).

## 2. Results and Discussion

### 2.1. Optical Properties of 4-Azidotriazoles ***1***

First, the photophysical data were obtained for family of 2-phenyl-5-aryl-4-azido-1,2,3-triazoles **1a**–**i** [30] having free rotation of the aryl(hetaryl) fragment in the 5 position of the triazole core (Figure 3a, Table 1). Compounds **1b**–**1i** exhibited bathochromic shifts in their absorption spectra relative to **1a** and showed strong maxima in the 311–346 nm region (Figure 4). The strongest bathochromic shift was observed for **1b** containing an electron-withdrawing NO_2_-group.

The emission spectra of compounds **1a**–**i** have two trends: spectra **1a**–**1d** and **1g**–**1i** exhibited similar emission bands in the 363–372 nm range. On the contrary, the emission spectra of pyridine analogues **1e** and **1f** demonstrated slightly different character having maxima at 419 nm and Stokes shifts up to 108 nm (Figure 5, Table 1). The reasons of such behavior are not clear at the moment and demand subsequent study, however it is most probably connected with the presence of additional nitrogen in the structure. Unfortunately, small fluorescence quantum yields were observed for all these compounds **1a**–**1i** (Table 1). 

### 2.2. Synthesis and Characterization of Condensed Analogues

Next, we decided to prepare some analogues of compounds **1a**–**1i** having a flattened structure and restricted rotation of the aryl fragment at the 5vposition of the triazole. We expected to enhance the fluorescence quantum yields by such a structural transformation. *ortho*-Propagyloxy- benzaldehydes were used as starting materials for this aim. A set of such aldehydes was prepared by alkylation of some salic aldehydes with propargyl bromide (Scheme 1).

The prepared *o*-propargyloxybenzaldehydes were converted into the corresponding 4,4-dichloro-1,2-diaza-1,3-dienes using the one-pot procedure previously elaborated by our team. This modification of the synthesis of 4,4-dichloro-1,2-diaza-1,3-dienes permits one to avoid isolation of hydrazones (Appendix A). 

All steps of the synthesis were performed in one pot to give the target dienes in respectable yields (up to 77%). It should be pointed out that the method is amenable to the variation of functional groups in the structure of the starting aldehydes. Electronically and sterically different *o*-propargyloxybenzaldehydes can be used for this aim. Moreover, the corresponding naphthalene derivative was prepared as well (Scheme 2 and Scheme 3) [26]. In a similar manner, a set of dienes **2h**–**o** having different aryl substituents at the nitrogen was prepared from the parent *o*-propargyloxy-benzaldehyde and substituted aryl hydrazines. We tried to perform variation of this part of molecules keeping in mind the influence of both electron and steric factors. For example, the corresponding dienes having methyl-, methoxy- and cyano groups, and different halogens can be prepared in up to 89% yield. Moreover, diene **2m** having a bulky 2,6-dimethylphenyl substituent was synthesized too.

Having in hand a family of precursors for the synthesis of model azidotriazoles, the reaction with excess of sodium azide was studied. It was found that the synthesis is very general to give the target products in up to 97% yield. A set of 15 triazoles **3a**–**o** was thus prepared having different substituents in the position 2 and 5 (Scheme 4) [30,32,33]. 

Next, intramolecular cyclization to form second triazole ring by thermal [2+3] cycloaddition was studied. Prepared compounds **3** have in the structure both an azide group and an acetylene fragment. We observed that spontaneous cyclization takes place slowly, even at room temperature, during storage. Smooth cyclization can be performed by reflux in *o*-xylene during 12 h in an argon atmosphere. As a result, a family of condensed triazole derivatives **4** having restricted rotation of substituent at the 5 position was prepared. We observed atropoisomerism [34] for some of prepared products. Their NMR spectra contain doubled set of signals (Scheme 5).

### 2.3. Photophysical Properties of Compounds ***4a***–***4o***

UV-vis absorption and fluorescence spectroscopic measurements were performed for the synthesized compounds **4** to establish the relationship between the structure and photophysical properties of the prepared flattened derivatives **4a**–**o** [35]. All these spectral data were obtained in dichloromethane (c = 10^−6^ M for all compounds) at room temperature and the results are summarized in Table 2. 

All the investigated compounds exhibited similar absorptions in the 250–305 nm range (Figure 6). The absorption spectra for compounds having a phenyl group at the N(2) position demonstrated absorption maxima at 284–288 nm. However, naphthalene derivative **4g** has a maximum of absorption shifted to 301 nm. More pronounced influence for absorption spectra was found when varying the substituents at N(2). The presence of an electron-withdrawing cyano group resulted in a bathochromic shift to 305 nm. On the other hand, a hypsochromic shift was observed for *ortho-*substituted derivatives, for example **4k**.

The emission spectra of solutions of **4a**–**4o** (Figure 7) were recorded at an excitation wavelength corresponding to the maximum in the absorption spectra. Typical emission maxima obtained upon irradiation of the solutions were located in the blue region. The Stokes shifts were shown in the range from 31 to 112 nm. 

Obviously, the spectral characteristics of compounds **4a**–**4o** depend on their electronic properties, conjugation of the substituents at the C(5) position and at the N(2) of the triazole. To our delight, much higher quantum yields (Φ_F_ up to 0.616) were observed for all derivatives **4a**–**4o** in comparison to triazole derivatives **1a**–**h** (Φ_F_ up to 0.017) having free rotation of the C(5) substituents. These data confirmed the attractiveness of our idea to synthesize and to study the photophysical properties of flattened intramolecular cyclizatioproducts. Analysis of the emission spectra for compounds **4a**–**4o** showed that the substitution at the N(2) position has more influence on fluorescence properties. A presence of a methoxy or cyano group at the aromatic ring *para*-position at the triazole N(2) leads to enhanced quantum yields (Table 2, Figure 6 and Figure 7), whereas, *ortho*-substituted derivatives exhibited lower fluorescenc efficiency. In contrast, the quantum yields were below 5% for any substituents at the C(5) of triazole ring (compounds **4a**–**f,** Φ_F_ up to 0.007–0.044). Due to the extra ring the naphthyl derivative **4g** showed moderate photophysical properties (Φ_F_ = 0.079).

The presence of a 4-MeO-group in the aryl at the N(2)-triazole resulted in the most effective conjugation, which resulted in a fluorescence enhancement (comp. **4l**, Φ_F_ up to 0.616). Most probably, better internal charge transfer (ICT) from the electron-donor OMe-group to the relatively electronically deficient part of molecule is achieved. On the contrary, the 2-methoxy derivative has the small quantum yield (comp. **4k**, Φ_F_ up to 0.013) but it gives the largest Stokes shift (112 nm), that makes it a pretty interesting fluorophore material. Other *ortho*-substituted at N(2)-triazole derivatives also demonstrated low quantum yields (**4i**, **4j**, **4m** and **4n)** to confirm the influence of steric hindrance on the photophysical properties. Most probably, lower conjugation of the aryl ring at the N(2)-triazole due to a distorted conformation is a reason for the reduced quantum yields in these cases. 

### 2.4. Solvatochromism of Compound 4l

We also carried out a study of solvatochromic properties for compound **4l** which demonstrated the highest quantum yields. The absorption and emission spectra of **4l** were taken at a standard concentration in different solvents of various polarities, including dioxane, benzene, EtOAc, THF, dichloromethane, EtOH, DMF, MeCN [36]. The UV-Vis spectrum of **4l** in low polarity benzene has a slight bathochromic shift relative to the spectrum in dichloromethane (Figure 8). Increasing solvent polarity resulted in more significant bathochromic shifts of the emission maxima, indicating an ICT behavior, which is better stabilized in polar solvents, for example in THF or EtOH (Figure 9). The intensity of the emission of compound **4l** is also highly dependent on the solvent polarity. In particular, the quantum yield of **4l** rises with increasing polarity of the solvents (benzene (0.457) < dioxane [0.497] < DMF [0.540] < EtOAc [0.583] < THF [0.591] < CH_2_Cl_2_ [0.616] < MeCN [0.618] < EtOH [0.670]) (Table 3). The lowest fluorescence quantum yield was observed in the nonpolar, aprotic solvent benzene because of the charge transfer phenomena [37]. The highest fluorescence quantum yield was observed in a polar, protic solvent, EtOH. 

## 3. Materials and Methods 

### 3.1. Experimental Details

All required fine chemicals were of reagent grade and were used directly without purification unless otherwise noted. ^1^H- and ^13^C-NMR spectra were acquired at 400.1 and 100.6 MHz, respectively, on an AVANCE 400 MHz spectrometer (Bruker, Karlsruhe, Germany) in chloroform-d (unless otherwise stated). ^1^H-NMR coupling constants (*J*) are reported in Hertz. Data are reported as follows: chemical shift, multiplicity (s - singlet, br s - broad singlet, d - doublet, t - triplet, q - quartet, m - multiplet, dd - doublet of doublets, ddd - doublet of double doublets), coupling constants, integration, and assignment (optionally). HRMS (ESI-MS) spectra were measured on MicroTof (Bruker Daltonics, Bremen, Germany). All IR data was obtained on a Nicolet iS5 One FT-IR spectrometer (Thermo Scientific, Madison, WI USA) using consoles of internal reflection iS3 with a ZnSe ATR element, dip angle 45 °C. All UV data was obtained on a Cary 60 UV-Visible spectrophotometer (Agilent, Santa Clara, CA USA) within 250–800 nm spectral range. The UV-spectra were recorded at 1 cm cuvettes at room temperature, dichloromethane was used as a solvent. Emission spectra were registered with a F2700 spectrofluorometer (Hitachi, Tokyo, Japan) in 1 cm quartz cells. The concentration of the compound in the solutions was 10^−5^ M and 10^−6^ M for both measurements. The relative fluorescence quantum yields (ΦF) were measured using quinine sulfate in 0.1 M H2SO4 (Φ_f_ = 0.55) and 2-aminopyridine 0.1 M H2SO4 (Φ_f_ = 0.60) as a standards [36]. 

Analytical TLC: aluminium-backed plates precoated (0.25 mm) with Silica Gel 60 F254 (Merck, Darmstadt, Germany) and 0.20 mm ALUGRAM^®^Xtra SIL G/UV_254_. (Macherey-Nagel, Düren, Germany). Compounds were visualized by exposure to UV light or by dipping the plates in permanganate (KMnO_4_) stain followed by heating. Flash and column chromatography were performed using Macherey-Nagel Silica gel 60 (70–230 mesh). All mixed solvent eluents are reported as v/v solutions. Solvents were purified by standard methods. DMSO was distilled over CaH_2_. Tetrachloromethane was distilled over P_2_O_5_.

### 3.2. General Procedure for the Preparation of (E)-1-(2,2-Dichloro-1-(2-(prop-2-yn-1-yloxy)phenyl)vinyl)-2-Phenyldiazenes

*o*-Propargyloxy-substituted dichlorodiazabutadienes were synthesized based on previous work [26]. A 20 mL screw neck vial was charged with DMSO (10 mL), the corresponding *o*-propargyloxy- benzaldehyde [31] (1 mmol, 1 eq.) and hydrazine (1 eq.). After 2 h stirring TMEDA (2.5 eq.), CuCl (0.01 eq.) were added and carbon tetrachloride (10 eq.) was put into the reaction mixture during 5 min under cooling with a water bath. The next reaction was carried out at room temperature during 3 h (until TLC analysis showed complete consumption of corresponding hydrazone). The reaction mixture was then poured into water (200 mL), and extracted with DCM (3 × 20 mL). The combined organic phase was washed with water (3 × 50 mL), brine (1 × 30 mL), dried over anhydrous sodium sulfate and concentrated in vacuo of the rotary evaporator. The residue was purified by column chromatography on silica gel using appropriate mixtures of hexane and DCM (3/1) as eluent.

*(E)-1-(2,2-Dichloro-1-(2-(prop-2-yn-1-yloxy)phenyl)vinyl)-2-phenyldiazene* (**2a**). Yield 248 mg (75%), orange-red solid, m.p. 85 °C. IR (ν, cm^−1^): 1573, 1584, 1604, 3261. ^1^H-NMR: δ 2.45 (t, 1H, *J* = 2.4 Hz), 4.63 (d, 2H, *J* = 2.4 Hz), 7.08–7.15 (m, 3H), 7.42–7.46 (m, 4H), 7.76–7.81 (m, 2H). ^13^C-NMR: δ 56.1, 75.5, 78.5, 112.7, 121.3, 122.9, 123.2, 128.9, 130.3, 131.3, 131.3, 136.0, 149.9, 153.0, 154.9. ESI-HRMS (*m/z*): calcd. for (C_17_H_13_Cl_2_N_2_O) [M + H^+^] 331.0400, found 331.0400.

*(E)-1-(2,2-Dichloro-1-(3-methoxy-2-(prop-2-yn-1-yloxy)phenyl)vinyl)-2-phenyldiazene* (**2b**). Yield 217 mg (60%), orange-red oil. IR (ν, cm^–1^): 1582, 1603, 3297. ^1^H-NMR: δ 2.33 (t, 1H, *J* = 2.5 Hz), 3.91 (s, 3H), 4.60 (s, 2H), 6.73 (dd, 1H, *J* = 7.7, 1.4 Hz), 7.01 (dd, 1H, *J* = 8.3, 1.3 Hz), 7.15 (t, 1H, *J* = 8.0 Hz), 7.42–7.45 (m, 3H), 7.76–7.80 (m, 2H). ^13^C-NMR: δ 55.7, 60.2, 74.9, 79.3, 113.2, 122.5, 123.2, 124.4, 128.0, 128.9, 131.3, 136.3, 144.7, 149.9, 152.4, 152.9. ESI-HRMS (*m/z*): calcd. for (C_187_H_15_Cl_2_N_2_O_2_) [M + H^+^] 361.0506, found 361.0505.

*(E)-1-(2,2-Dichloro-1-(3-nitro-2-(prop-2-yn-1-yloxy)phenyl)vinyl)-2-phenyldiazene* (**2c**). Yield 196 mg (52%), orange oil. IR (ν, cm^–1^): 1533, 1574, 1602, 3292. ^1^H-NMR: δ 2.49 (t, 1H, *J* = 2.5 Hz), 4.55 (d, 2H, *J* = 9.5 Hz), 7.32–7.37 (m, 2H), 7.43–7.49 (m, 3H), 7.75–7.79 (m, 2H), 7.98 (dd, 1H, *J* = 7.0, 2.9 Hz). ^13^C-NMR: δ 62.6, 76.8, 77.4, 110.5, 123.3, 124.6, 126.3, 129.1, 130.9, 132.0, 133.3, 136.3, 137.6, 144.3, 148.2, 149.5, 152.5. ESI-HRMS (*m/z*): calcd. for (C_17_H_12_Cl_2_N_3_O_3_) [M + H^+^] 376.0251, found 376.0275.

*(E)-1-(2,2-Dichloro-1-(5-nitro-2-(prop-2-yn-1-yloxy)phenyl)vinyl)-2-phenyldiazene* (**2d**). Yield 290 mg (77%), orange-red solid, m.p. 105 °C. IR (ν, cm^−1^): 1568, 1585, 1612, 3290. ^1^H-NMR: δ 2.53 (t, 1H, *J* = 2.4 Hz), 4.74 (d, 2H, *J* = 2.4 Hz), 7.20 (d, 1H, *J* = 9.2 Hz), 7.41–7.48 (m, 3H), 7.72–7.77 (m, 2H), 8.06 (d, 1H, *J* = 2.8 Hz), 8.35 (dd, 1H, *J* = 9.2, 2.8 Hz). ^13^C-NMR: δ 56.7, 77.0, 77.5, 112.3, 123.4, 123.9, 126.6, 127.6, 129.2, 131.9, 137.2, 141.9, 148.1, 152.8, 159.7. ESI-HRMS (*m*/z): calcd. for (C_17_H_12_Cl_2_N_3_O_3_) [M + H^+^] 376.0251, found 376.0260.

*(E)-1-(1-(5-Bromo-2-(prop-2-yn-1-yloxy)phenyl)-2,2-dichlorovinyl)-2-phenyldiazene* (**2e**). Yield 271 mg (66%), orange-red solid, m.p. 107 °C. IR (ν, cm^−1^): 1564, 1582, 1598, 3310. ^1^H-NMR: δ 2.46 (t, 1H, *J* = 2.4 Hz), 4.60 (d, 2H, *J* = 2.4 Hz), 7.01 (d, 1H, *J* = 8.8 Hz), 7.26 (d, 1H, *J* = 2.5 Hz), 7.44–7.47 (m, 3H), 7.52 (dd, 1H, *J* = 8.9, 2.5 Hz), 7.76–7.80 (m, 2H). ^13^C-NMR: δ 56.2, 75.9, 77.9, 113.4, 114.4, 123.2, 124.9, 128.9, 131.5, 133.0, 133.8, 136.5, 148.6, 152.8, 154.0. ESI-HRMS (*m/z*): calcd. for (C_17_H_12_BrCl_2_N_2_O) [M + H^+^] 408.9505, found 408.9520.

*(E)-1-(2,2-Dichloro-1-(3,5-di-tert-butyl-2-(prop-2-yn-1-yloxy)phenyl)vinyl)-2-phenyldiazene* (**2f**). Yield 230 mg (52%), orange-red oil. IR (ν, cm^−1^): 1562, 1599, 3307. ^1^H-NMR: δ 1.31 (s, 9H), 1.45 (s, 9H), 2.50 (t, 1H, *J=*2.5 Hz), 4.19–4.38 (m, 2H), 6.87 (d, 1H, *J* = 2.5 Hz), 7.41 (d, 1H, *J* = 2.5 Hz), 7.45–7.47 (m, 3H), 7.79–7.81 (m, 2H). ^13^C-NMR: δ 30.3, 31.0, 34.2, 34.8, 60.6, 74.6, 78.7, 122.9, 124.5, 125.0, 126.3, 128.6, 131.1, 136.4, 141.4, 145.1, 150.7, 152.6, 154.0. ESI-HRMS (*m/z*): calcd. for (C_25_H_29_Cl_2_N_2_O) [M + H^+^] 443.1652, found 443.1649.

*(E)-1-(2,2-Dichloro-1-(2-(prop-2-yn-1-yloxy)naphthalen-1-yl)vinyl)-2-phenyldiazene* (**2g**). Yield 175 mg (46%), orange-red oil. IR (ν, cm^−1^): 1567, 1587, 1595, 3295. ^1^H-NMR: δ 2.45 (t, 1H, *J* = 2.4 Hz), 4.75 (d, 2H, *J* = 2.4 Hz), 7.36–7.44 (m, 6H), 7.47 (d, 1H, *J* = 9.1 Hz), 7.70–7.73 (m, 2H), 7.85 (d, 1H, *J* = 7.7 Hz), 7.96 (d, 1H, *J* = 9.1 Hz). ^13^C-NMR: δ 57.0, 75.6, 78.8, 114.6, 117.9, 123.2, 124.1, 124.2, 127.2, 128.2, 128.8, 129.2, 131.0, 131.3, 132.1 136.9, 148.2, 152.7, 153.0. ESI-HRMS (*m/z*): calcd. for (C_21_H_15_Cl_2_N_2_O) [M + H^+^] 381.0556, found 381.0564.

*(E)-1-(2,2-Dichloro-1-(2-(prop-2-yn-1-yloxy)phenyl)vinyl)-2-(4-fluorophenyl)diazene* (**2h**). Yield 175 mg (50%), orange-red solid, m.p. 60 °C. IR (ν, cm^-1^): 1574, 1592, 1603, 3301. ^1^H-NMR: δ 2.44 (t, 1H, *J* = 2.4 Hz), 4.63 (d, 2H, *J* = 2.4 Hz), 7.06–7.13 (m, 5H), 7.41–7.45 (m, 1H), 7.76–7.82 (m, 2H). ^13^C-NMR: δ 56.1, 75.5, 78.4, 112.7, 115.8, 116.0, 121.3, 122.8, 125.2, 125.3, 130.4, 131.3, 149.5, 149.7, 154.9, 163.3, 165.8. ESI-HRMS (*m/z*): calcd. for (C_17_H_12_Cl_2_FN_2_O) [M + H^+^] 349.0305, found 349.0307.

*(E)-1-(2,2-Dichloro-1-(2-(prop-2-yn-1-yloxy)phenyl)vinyl)-2-(2,4-dichlorophenyl)diazene* (**2i**). Yield 244 mg (61%), orange-red solid, m.p. 104 °C. IR (ν, cm^−1^): 1575, 1599, 3300. ^1^H-NMR: δ 2.44 (t, 1H, *J* = 2.4 Hz), 4.62 (d, 2H, *J* = 2.3 Hz), 7.05–7.15 (m, 3H), 7.27–7.29 (m, 1H), 7.40–7.44 (m, 1H), 7.47 (d, 1H, *J* = 2.1 Hz), 7.63 (d, 1H, *J* = 8.7 Hz). ^13^C-NMR: δ 56.3, 75.5, 78.4, 112.5, 118.5, 121.3, 122.3, 127.6, 130.3, 130.6, 131.2, 136.5, 137.4, 137.9, 147.6, 150.8, 155.1. ESI-HRMS (*m/z*): calcd. for (C_17_H_11_Cl_4_N_2_O) [M + H^+^] 398.9620, found 398.9622.

*(E)-1-(2-Chlorophenyl)-2-(2,2-dichloro-1-(2-(prop-2-yn-1-yloxy)phenyl)vinyl)diazene* (**2j**). Yield 201 mg (55%), orange-red oil. IR (ν, cm^−1^): 1581, 1602, 3297. ^1^H-NMR: δ 2.44 (t, 1H, *J* = 2.4 Hz), 4.63 (d, 2H, *J* = 2.3 Hz), 7.05–7.15 (m, 3H), 7.30–7.38 (m, 2H), 7.39-7.45 (m, 2H), 7.65 (dd, 1H, *J* = 7.8, 2.0 Hz). ^13^C-NMR: δ 56.4, 75.5, 78.6, 112.6, 117.7, 121.3, 122.5, 127.1, 130.5, 130.6, 131.2, 131.9, 135.7, 137.3, 149.1, 150.6, 155.2. ESI-HRMS (*m/z*): calcd. for (C_17_H_12_Cl_3_N_2_O) [M + H^+^] 365.0010, found 365.0005.

*(E)-1-(2,2-Dichloro-1-(2-(prop-2-yn-1-yloxy)phenyl)vinyl)-2-(2-methoxyphenyl)diazene* (**2k**). Yield 235 mg (65%), orange-red solid, m.p. 102 °C. IR (ν, cm^−1^): 1570, 1585, 1594, 1607, 3235. ^1^H-NMR: δ 2.44 (t, 1H, *J* = 2.4 Hz), 3.73 (s, 3H), 4.61 (d, 2H, *J* = 2.5 Hz), 6.95–7.00 (m, 2H), 7.07 (td, 1H, *J* = 7.4, 0.9 Hz), 7.14–7.16 (m, 2H), 7.35–7.42 (m, 2H), 7.62 (dd, 1H, *J* = 7.9, 1.6 Hz). ^13^C-NMR: δ 56.6, 57.2, 75.3, 78.7, 113.1, 114.3, 117.3, 120.9, 121.5, 123.1, 130.3, 131.3, 132.7, 135.2, 143.0, 150.3, 155.3, 157.3. ESI-HRMS (*m/z*): calcd. for (C_18_H_15_Cl_2_N_2_O_2_) [M + H^+^] 361.0505, found 361.0519.

*(E)-1-(2,2-Dichloro-1-(2-(prop-2-yn-1-yloxy)phenyl)vinyl)-2-(4-methoxyphenyl)diazene* (**2l**). Yield 101 mg (28%), orange-red solid, m.p. 69 °C. IR (ν, cm^−1^): 1569, 1581, 1601, 3286. ^1^H-NMR: δ 2.43 (t, 1H, *J* = 2.4 Hz), 3.86 (s, 3H), 4.62 (d, 2H, *J* = 2.4 Hz), 6.92 (dt, 2H, *J* = 9.8, 2.6 Hz), 7.07 (td, 1H, *J* = 7.3, 0.9 Hz), 7.10–7.14 (m, 2H), 7.39–7.44 (m, 1H), 7.77 (dt, 2H, *J* = 9.7, 2.6 Hz). ^13^C-NMR: δ 55.5, 56.2, 75.4, 78.6, 112.8, 114.0, 121.3, 123.3, 125.2, 130.2, 131.4, 133.9, 147.4, 149.7, 155.0, 162.3. ESI-HRMS (*m/z*): calcd. for (C_18_H_15_Cl_2_N_2_O_2_) [M + H^+^] 361.0505, found 361.0512.

*(E)-1-(2,2-Dichloro-1-(2-(prop-2-yn-1-yloxy)phenyl)vinyl)-2-(2,4-dimethylphenyl)diazene* (**2m**). Yield 320 mg (89%), orange-red oil. IR (ν, cm^−1^): 1575, 1584, 1606, 3298. ^1^H-NMR: δ 2.24 (s, 3H), 2.35 (s, 3H), 2.44 (t, 1H, *J* = 2.4 Hz), 4.62 (d, 2H, *J* = 2.4 Hz), 7.03–7.13 (m, 5H), 7.39–7.43 (m, 1H), 7.57–7.60 (m, 1H). ^13^C-NMR: δ 17.0, 21.4, 56.1, 75.5, 78.5, 112.4, 115.4, 121.1, 123.6, 127.1, 130.1, 131.2, 131.7, 134.2, 139.0, 141.9, 148.9, 150.4, 154.9. ESI-HRMS (*m/z*): calcd. for (C_19_H_17_Cl_2_N_2_O) [M + H^+^] 359.0712, found 359.0715.

*(E)-1-(2,2-Dichloro-1-(2-(prop-2-yn-1-yloxy)phenyl)vinyl)-2-(2,6-dimethylphenyl)diazene* (**2n**). Yield 320 mg (89%), orange-red oil. IR (ν, cm^−1^): 1584, 1605, 3300. ^1^H-NMR: δ 2.29 (s, 6H), 2.47 (t, 1H, *J* = 2.4 Hz), 4.65 (s, 2H), 7.09–7.18 (m, 6H), 7.42–7.46 (m, 1H). ^13^C-NMR: δ 19.4, 56.0, 75.6, 78.3, 112.2, 121.2, 123.1, 129.0, 129.1, 130.2, 131.1, 135.5, 150.5, 150.6, 154.7. ESI-HRMS (*m/z*): calcd. for (C_19_H_17_Cl_2_N_2_O) [M + H^+^] 359.0712, found 359.0718.

*(E)-4-((2,2-Dichloro-1-(2-(prop-2-yn-1-yloxy)phenyl)vinyl)diazenyl)benzonitrile* (**2o**). Yield 146 mg (41%), orange-red solid, m.p. 154 °C. IR (ν, cm^−1^): 1578, 1608, 3303. ^1^H-NMR: δ 2.44 (t, 1H, *J* = 2.4 Hz), 4.63 (d, 2H, *J* = 2.3 Hz), 7.07–7.13 (m, 3H), 7.42–7.47 (m, 1H), 7.71–7.73 (m, 2H), 7.82–7.84 (m, 2H). ^13^C-NMR: δ 55.9, 75.6, 78.2, 112.5, 114.0, 118.4, 121.4, 122.0, 123.6, 130.6, 131.2, 133.0, 139.3, 150.3, 154.7, 154.8. ESI-HRMS (*m/*z): calcd. for (C_18_H_12_Cl_2_N_3_O) [M + H^+^] 356.0352, found 356.0347.

### 3.3. General Procedure for Preparation of 4-Azido-2-phenyl-5-(2-(prop-2-yn-1-yloxy)phenyl)-2H-1,2,3-triazoles

Attention! All manipulations with any azides demand significant care due to safety reasons!

*o*-Propargyloxy-substituted 4-azido-2,5-diaryl-1,2,3-triazoles were obtained according to the procedure described in this paper [27]. A 20 mL screw neck vial was charged with DMSO (10 mL) and sodium azide (5 eq.), then corresponding dichlorodiazene (1 mmol, 1 eq.) was added dropwise over 15 min. The resulting mixture was stirred at room temperature for 2 h. Next, reaction mixture was poured into water (100 mL), extracted with DCM (3 × 20 mL). Combined extract was washed with water (3 × 50 mL), brine (1 × 30 mL) and dried over sodium sulfate. Volatiles were removed in vacuum of the rotary evaporator, the residue was purified by column chromatography on silica gel using a mixture of hexane and DCM (3/1). Compounds **3h**, **3k**, **3l**, **3m** and **3n** were obtained as a mixture of two atropoisomers. Thus, the NMR spectra of these compounds have dual signals.

*4-Azido-2-phenyl-5-(2-(prop-2-yn-1-yloxy)phenyl)-2H-1,2,3-triazole* (**3a**). Yield 138 mg (64%), yellow oil. IR (ν, cm^-1^): 3290, 2131. ^1^H-NMR: δ 2.55 (t, 1H, *J* = 2.4 Hz), 4.80 (d, 2H, *J* = 2.4 Hz), 7.11–7.17 (m, 2H), 7.29–7.36 (m, 1H), 7.42–7.51 (m, 4H), 8.06-8.08 (m, 2H). ^13^C-NMR: δ 56.1, 75.5, 78.5, 112.7, 121.3, 122.9, 123.2, 128.9, 130.3, 131.3, 131.3, 136.0, 149.9, 153.0, 154.9. ESI-HRMS (*m/z*): calcd. for (C_17_H_13_N_6_O) [M + H^+^] 317.1146, found 317.1146.

*4-Azido-5-(3-methoxy-2-(prop-2-yn-1-yloxy)phenyl)-2-phenyl-2H-1,2,3-triazole* (**3b**). Yield 43 mg (68%), white solid, m. p. 102 °C. IR (ν, cm^−1^): 3296, 2125. ^1^H-NMR: δ 2.42 (t, 1H, *J* = 2.4 Hz), 3.92 (s, 3H), 4.82 (d, 2H, *J* = 2.4 Hz), 7.01–7.03 (m, 1H), 7.17–7.21 (m, 2H), 7.35 (t, 1H, *J* = 7.4 Hz), 7.49 (t, 2H, *J* = 7.9 Hz), 8.10 (d, 2H*, J* = 7.8 Hz). ^13^C-NMR: δ 55.9, 60.6, 74.8, 79.2, 113.3, 118.2, 122.1, 123.6, 124.8, 127.2, 129.2, 135.9, 139.5, 144.3, 144.8, 153.2. ESI-HRMS (*m/z*): calcd. for (C_18_H_15_N_6_O_2_) [M + H^+^] 347.1251, found 347.0787.

*4-Azido-5-(3-nitro-2-(prop-2-yn-1-yloxy)phenyl)-2-phenyl-2H-1,2,3-triazole* (**3c**). Yield 114 mg (63%), yellow solid, m. p. 120 °C. IR (ν, cm^−1^): 3303, 2130. ^1^H-NMR: δ 2.46 (t, 1H, *J* = 2.4 Hz), 4.78 (d, 2H*, J* = 2.5 Hz), 7.36–7.41 (m, 2H), 7.51 (t, 2H, *J* = 7.9 Hz), 7.91 (d, 2H, *J* = 8.2 Hz), 8.10 (d, 2H, *J* = 8.2 Hz). ^13^C-NMR: δ 63.1, 76.7, 77.5, 118.3, 125.0, 125.8, 126.1, 127.8, 129.4, 133.6, 135.2, 139.2, 144.7, 145.5, 148.8. ESI-HRMS (*m/z*): calcd. for (C_17_H_12_BrN_6_O) [M + H^+^] 395.0251, found 395.0250.

*4-Azido-5-(5-nitro-2-(prop-2-yn-1-yloxy)phenyl)-2-phenyl-2H-1,2,3-triazole* (**3d**). Yield 114 mg (63%), white-yellow solid, m. p. 160 °C. IR (ν, cm^−1^): 3286, 2131. ^1^H-NMR: δ 2.57 (t, 1H, *J* = 2.4 Hz), 4.78 (d, 2H, *J* = 2.4 Hz), 7.04 (d, 1H, *J* = 8.8 Hz), 7.34–7.37 (m, 1H), 7.47–7.54 (m, 3H), 7.72 (d, 1H, *J* = 2.5 Hz), 8.05–8.07 (m, 2H). ^13^C-NMR: δ 58.9, 76.1, 77.8, 114.0, 115.1, 118.2, 120.5, 127.4, 129.3, 133.1, 133.6, 134.8, 139.3, 144.4, 154.3. ESI-HRMS (*m/z*): calcd. for (C_17_H_12_BrN_6_O) [M + H^+^] 395.0251, found 395.0250.

*4****-****Azido-5-(5-bromo-2-(prop-2-yn-1-yloxy)phenyl)-2-phenyl-2H-1,2,3-triazole* (**3e**). Yield 158 mg (80%), yellow solid, m.p. 218 °C. IR (ν, cm^−1^): 3292, 2131. ^1^H-NMR: δ 2.57 (t, 1H, *J* = 2.4 Hz), 4.78 (d, 2H, *J* = 2.4 Hz), 7.04 (d, 1H, *J* = 8.8 Hz), 7.34–7.37 (m, 1H), 7.47–7.54 (m, 3H), 7.72 (d, 1H, *J* = 2.5 Hz), 8.05–8.07 (m, 2H). ^13^C-NMR: δ 58.9, 76.1, 77.8, 114.0, 115.1, 118.2, 120.5, 127.4, 129.3, 133.1, 133.6, 134.8, 139.3, 144.4, 154.3. ESI-HRMS (*m/z*): calcd. for (C_17_H_12_BrN_6_O) [M + H^+^] 395.0251, found 395.0250.

*4-Azido-5-(3,5-di-tert-butyl-2-(prop-2-yn-1-yloxy)phenyl)-2-phenyl-2H-1,2,3-triazole* (**3f**). Yield 231 mg (59%), white-yellow solid, m.p. 114 °C. IR (ν, cm^−1^): 3310, 2131. ^1^H-NMR: δ 1.38 (s, 9H), 1.51 (s, 9H), 2.42 (t, 1H, *J* = 2.4 Hz), 4.28 (dd, 2H, *J* = 14.9, 2.4 Hz), 7.35–7.39 (m, 2H), 7.49–7.53 (m, 3H), 8.13 (d, 2H, *J* = 7.7 Hz). ^13^C-NMR: δ 31.0, 31.4, 34.6, 35.3, 61.2, 74.9, 78.7, 118.2, 118.5, 125.9, 126.6, 127.3, 127.8, 129.2, 137.5, 139.4, 142.7, 144.2, 146.4, 153.8. ESI-HRMS (*m/z*): calcd. for (C_25_H_29_N_6_O) [M + H^+^] 429.2397, found 429.2410.

*4-Azido-2-phenyl-5-(2-(prop-2-yn-1-yloxy)naphthalen-1-yl)-2H-1,2,3-triazole* (**3g**). Yield 38 mg (35%), yellow oil. IR (ν, cm^−1^): 2130. ^1^H-NMR: δ 2.52 (t, 1H, *J* = 2.5 Hz), 4.84 (d, 2H, *J* = 2.4 Hz), 7.37–7.55 (m, 6H), 7.66–7.68 (m, 1H), 7.88–7.90 (m, 1H), 8.03 (d, 1H, *J* = 9.1 Hz), 8.13-8.16 (m, 2H). ^13^C-NMR: δ 57.2, 76.0, 78.4, 111.8, 114.5, 118.4, 124.4, 127.4, 127.7, 128.2, 129.3, 131.7, 133.4, 139.4, 139.6, 154.1. ESI-HRMS (*m/z*): calcd. for (C_21_H_15_N_6_O) [M + H^+^] 367.1302, found 367.1309.

*4-Azido-2-(4-fluorophenyl)-5-(2-(prop-2-yn-1-yloxy)phenyl)-2H-1,2,3-triazole* (**3h**). Yield 155 mg (93%), yellow oil. IR (ν, cm^−1^): 2131. ^1^H-NMR: δ 2.57 (t, 2H, *J* = 2.3 Hz), 4.80 (d, 4H, *J* = 2.4 Hz), 7.11–7.21 (m, 8H), 7.42–7.51 (m, 2H), 7.54–7.58 (m, 2H), 8.01–8.07 (m, 2H). ^13^C-NMR: δ 56.1, 56.5, 75.7, 75.8, 78.2, 78.2, 112.9, 113.2, 115.9, 115.9, 116.1, 116.2, 117.9, 118.1, 119.7, 119.8, 120.0, 120.1, 121.5, 121.7, 130.7, 130.9, 131.1, 131.4, 135.7, 135.7, 136.3, 144.3, 155.2, 155.3, 160.3, 160.5, 162.7, 163.0. ESI-HRMS (*m/z*): calcd. for (C_17_H_12_FN_6_O) [M + H^+^] 335.1052, found 335.1053.

*4-Azido-2-(2,4-dichlorophenyl)-5-(2-(prop-2-yn-1-yloxy)phenyl)-2H-1,2,3-triazole* (**3i**). Yield 121 mg (63%), yellow oil. IR (ν, cm^−1^): 3298, 2131. ^1^H-NMR: δ 2.55 (t, 1H, *J* = 2.4 Hz), 4.80 (d, 2H, *J* = 2.4 Hz), 7.10–7.17 (m, 2H), 7.38–7.46 (m, 2H), 7.56 (dd, 1H, *J* = 7.6, 1.7 Hz), 7.60 (d, 1H, *J* = 2.3 Hz), 7.66 (d, 1H, *J* = 8.6 Hz). ^13^C-NMR (100 MHz, CDCl_3_): δ 56.7, 75.8, 78.3, 113.3, 118.0, 121.8, 127.7, 127.8, 129.6, 130.8, 131.0, 131.2, 135.1, 136.3, 136.9, 144.8, 155.3. ESI-HRMS (*m/z*): calcd. for (C_17_H_11_Cl_2_N_6_O) [M + H^+^] 385.0366, found 385.0363.

*4-Azido-2-(2-chlorophenyl)-5-(2-(prop-2-yn-1-yloxy)phenyl)-2H-1,2,3-triazole* (**3j**). Yield 109 mg (62%), yellow solid, m.p. 162 °C. IR (ν, cm^−1^): 3296, 2128. ^1^H-NMR: δ 2.55 (t, 1H, *J* = 2.4 Hz), 4.81 (d, 2H, *J* = 2.4 Hz), 7.10–7.18 (m, 2H), 7.38–7.46 (m, 3H), 7.56–7.59 (m, 2H), 7.69–7.71 (m, 1H). ^13^C-NMR: δ 56.7, 75.7, 78.4, 113.4, 118.2, 121.8, 127.3, 127.4, 129.1, 130.0, 130.7, 131.1, 131.3, 136.5, 137.7, 144.5, 155.3. ESI-HRMS (*m*/*z*): calcd. for (C_17_H_12_ClN_6_O) [M + H^+^] 351.0756, found 351.0754.

*4-Azido-2-(2-methoxyphenyl)-5-(2-(prop-2-yn-1-yloxy)phenyl)-2H-1,2,3-triazole* (**3k**). Yield 149 mg (86%), yellow oil. IR (ν, cm^−1^): 3292, 2132. ^1^H-NMR: δ 2.55 (t, 2H, *J* = 2.4 Hz), 3.90 (s, 6H), 4.79 (d, 4H*, J* = 2.5 HZ), 7.05–7.20 (m, 8H), 7.39–7.48 (m, 4H), 7.55–7.60 (m, 4H). ^13^C-NMR: δ 56.2, 56.3, 75.7, 75.7, 78.4, 78.4, 112.7, 112.8, 113.0, 113.3, 118.2, 118.5, 120.5, 120.6, 121.5, 121.7, 126.9, 129.4, 129.5, 130.4, 130.4, 130.6, 130.7, 131.3, 131.6, 135.9, 137.1, 142.7, 143.8, 153.3, 153.4, 155.3, 155.4. ESI-HRMS (*m/z*): calcd. for (C_18_H_15_N_6_O_2_) [M + H^+^] 347.1251, found 347.1257. 

*4-Azido-2-(4-methoxyphenyl)-5-(2-(prop-2-yn-1-yloxy)phenyl)-2H-1,2,3-triazole* (**3l**). Yield 102 mg (59%), yellow oil. IR (ν, cm^−1^): 3292, 2131. ^1^H-NMR: δ 2.55 (t, 1H, *J* = 2.3 Hz), 2.57 (t, 1H, *J* = 2.3 Hz), 3.83 (s, 6H), 4.78 (d, 2H, *J* = 2.4 Hz), 4.79 (d, 2H, *J* = 2.4 Hz), 6.97–7.01 (m, 4H), 7.10–7.20 (m, 4H), 7.40–7.49 (m, 2H), 7.55–7.60 (m, 2H), 7.96–8.01 (m, 4H). ^13^C-NMR: δ 55.4, 56.1, 56.5, 75.6, 75.7, 78.2, 78.3, 112.9, 113.2, 114.2, 114.2, 118.1, 118.4, 119.5, 119.7, 121.4, 121.6, 130.4, 130.7, 131.0, 131.3, 133.1, 133.2, 133.7, 134.0, 135.5, 137.0, 142.5, 143.6, 155.2, 155.3, 158.7, 158.9. ESI-HRMS (*m/z*): calcd. for (C_18_H_15_N_6_O_2_) [M + H^+^] 347.1251, found 347.1251.

*4-Azido-2-(2,4-dimethylphenyl)-5-(2-(prop-2-yn-1-yloxy)phenyl)-2H-1,2,3-triazole* (**3m**). Yield 162 mg (56%), yellow oil. IR (ν, cm^−1^): 3292, 2119. ^1^H-NMR: δ 2.39 (s, H), 2.45–2.47 (m, 6H), 2.53 (t, 1H, *J* = 2.4 Hz), 2.55 (t, 1H, *J* = 2.4 Hz), 4.79 (d, 2H, *J* = 2.4 Hz), 4.80 (d, 2H, *J* = 2.5 Hz), 7.09–7.21 (m, 8H), 7.40–7.49 (m, 2H), 7.51–7.58 (m, 4H). ^13^C-NMR: δ 18.9, 19.1, 21.0, 21.0, 56.3, 56.7, 75.6, 75.7, 78.3, 78.4, 113.1, 113.3, 118.3, 118.6, 121.5, 121.7, 124.6, 124.8, 127.2, 130.4, 130.7, 131.2, 131.5, 132.0, 132.2, 132.3, 132.3, 136.7, 137.0, 138.5, 138.9, 142.3, 143.5, 155.3, 155.4. ESI-HRMS (*m/z*): calcd. for (C_19_H_17_N_6_O) [M + H^+^] 345.1459, found 345.1458. 

*4-Azido-2-(2,6-dimethylphenyl)-5-(2-(prop-2-yn-1-yloxy)phenyl)-2H-1,2,3-triazole* (**3n**). Yield 168 mg (58%), yellow oil. IR (ν, cm^−1^): 3291, 2126. ^1^H-NMR: δ 2.14 (s, 6H), 2.16 (s, 6H), 2.52 (t, 1H, *J* = 2.4 Hz), 2.53 (t, 1H, *J* = 2.4 Hz), 4.78 (dd, 2H, *J* = 2.5 Hz), 4.79 (dd, 2H, *J* = 2.5 Hz), 7.09–7.20 (m, 8H), 7.28–7.33 (m, 2H), 7.40–7.49 (m, 2H), 7.56 (t, 1H, *J* = 1.7 Hz), 7.58 (t, 1H, *J* = 1.7 Hz). ^13^C-NMR: δ 17.4, 17.5, 56.4, 56.7, 75.6, 75.7, 78.3, 78.4, 113.2, 113.4, 118.3, 118.5, 121.5, 121.7, 128.2, 128.3, 129.8, 129.9, 130.4, 130.7, 131.1, 131.4, 135.1, 135.9, 136.0, 136.4, 139.1, 139.2, 142.1, 143.4, 155.3, 155.4. ESI-HRMS (*m/z*): calcd. for (C_19_H_17_N_6_O) [M + H^+^] 345.1459, found 345.1459. 

*4-(4-Azido-5-(2-(prop-2-yn-1-yloxy)phenyl)-2H-1,2,3-triazol-2-yl)benzonitrile* (**3o**). Yield 102 mg (60%), grey solid, m. p. 124 °C. IR (ν, cm^−1^): 3256, 2133. ^1^H-NMR: δ 2.56 (t, 1H, *J* = 2.4 Hz), 4.80 (d, 2H, *J* = 2.4 Hz), 7.11–7.18 (m, 2H), 7.45–7.49 (m, 1H), 7.56 (dd, 1H, *J* = 7.6, 1.7 Hz), 7.77 (dt, 2H, *J* = 9.0, 2.0 Hz), 8.17 (dt, 2H, *J* = 9.0, 2.0 Hz). ^13^C-NMR: δ 56.6, 75.8, 78.1, 110.3, 113.2, 117.6, 118.3, 118.3, 121.8, 131.1, 131.2, 133.5, 138.0, 141.9, 145.9, 155.4. ESI-HRMS (*m/z*): calcd. for (C_18_H_12_N_7_O) [M + H^+^] 342.1098, found 342.1099. 

### 3.4. General Procedure for Preparation of 2-Aryl-2H,8H-benzo[g]bis([1,2,3]triazolo)[5,1-c:4’,5’-e][1,4]-oxazocines ***4***

To 3 mL of argon-sparged *o*-xylene the previously synthesized 4-azido-2-phenyl-5-(2-(prop-2-yn-1-yloxy)phenyl)-2*H*-1,2,3-triazole was added. The reaction mixture was refluxed under Ar for 12 h or more until TLC analysis showed complete consumption of source reagent. The final reaction mixture was evaporate and the residue was purified by column chromatography on silica gel using a mixture of hexane and EtOAc (6/1) as eluent.

*2-Phenyl-2H,8H-benzo[g]bis(**[1,2,3]**triazolo)[5,1-c:4’,5’-e]**[1,4]**oxazocine* (**4a**). Yield 222 mg (70%), white solid, m. p. 157 °C. IR (ν, cm^−1^): no signals of azido group and triple bond. ^1^H-NMR: δ 5.58 (s, 2H), 7.26–7.30 (m, 2H), 7.40 (t, 1H, *J* = 7.4 Hz), 7.45–7.54 (m, 3H), 7.63 (s, 1H), 7.71 (dd, 1H, *J* = 8.1, 1.6 Hz), 8.19–8.21 (m, 2H). ^13^C NMR (100 MHz, CDCl_3_): δ 67.0, 118.8, 121.4, 121.6, 125.4, 128.3, 129.3, 129.9, 131.6, 131.8, 134.7, 138.3, 139.1, 141.6, 155.6. ESI-HRMS (*m*/*z*): calcd. for (C_17_H_13_N_6_O) [M + H^+^] 317.1145, found 317.1144.

*10-Methoxy-2-phenyl-2H,8H-benzo[g]bis(**[1,2,3]**triazolo)[5,1-c:4’,5’-e]**[1,4]**oxazocine* (**4b**). Yield 291 mg (84%), white-yellow solid, m. p. 170 °C. IR (ν, cm^−1^): no signals of azido group and triple bond were observed. ^1^H-NMR: δ 3.95 (s, 3H), 5.65 (s, 2H), 7.02–7.06 (m, 1H), 7.22–7.25 (m, 2H), 7.42 (tt, 1H, *J* = 7.4, 1.3 Hz), 7.51–7.55 (m, 2H), 7.62 (s, 1H), 8.19–8.23 (m, 2H). ^13^C-NMR: δ 55.9, 65.4, 113.5, 118.9, 120.8, 123.6, 126.2, 128.2, 129.3, 131.5, 135.5, 138.4, 139.3, 142.2, 143.7, 152.6. ESI-HRMS (*m/z*): calcd. for (C_18_H_15_N_6_O) [M + H^+^] 347.1251, found 347.1247.

*10-Nitro-2-phenyl-2H,8H-benzo[g]bis(**[1,2,3]**triazolo)[5,1-c:4’,5’-e]**[1,4]**oxazocine* (**4c**). Yield 264 mg (73%), white solid, m. p. 250 °C. IR (ν, cm^−1^): no signals of azido group and triple bond were observed. ^1^H-NMR (DMSO-d_6_): δ 5.83 (s, 2H), 7.55–7.59 (m, 2H), 7.68 (t, 2H, *J* = 7.9 Hz), 8.03 (s, 1H), 8.17 (d, 2H, *J* = 7.9 Hz), 8.23 (d, 2H, *J* = 8.0 Hz). ^13^C-NMR (DMSO-d_6_): δ 66.2, 118.9, 123.8, 125.9, 127.6, 129.2, 130.1, 133.3, 134.7, 135.7, 136.6, 138.5, 140.9, 143.5, 148.3. ESI-HRMS (*m/z*): calcd. for (C_17_H_12_N_7_O_3_) [M + H^+^] 362.0996, found 362.0997.

*12-Nitro-2-phenyl-2H,8H-benzo[g]bis(**[1,2,3]**triazolo)[5,1-c:4’,5’-e]**[1,4]**oxazocine* (**4d**). Yield 246 mg (68%), white solid, m. p. 237 °C. IR (ν, cm^−1^): no signals of azido group and triple bond were observed. ^1^H-NMR: δ 5.58 (s, 2H), 7.34 (d, 1H, *J* = 9.1 Hz), 7.47–7.51 (m, 1H), 7.58 (t, 2H, *J* = 7.8 Hz), 7.86 (s, 1H), 8.23–8.25 (m, 2H), 8.29 (dd,1H, *J* = 9.1, 2.8 Hz), 8.78 (d, 1H, *J* = 2.8 Hz). ^13^C-NMR: δ 63.5, 110.6, 119.1, 122.9, 126.4, 127.6, 129.1, 129.6, 133.0, 133.3, 136.7, 138.9, 144.0, 159.8, 160.6, 169.7, 178.8. ESI-HRMS (*m/z*): calcd. for (C_17_H_12_N_7_O_3_) [M + H^+^] 362.0996, found 362.1008.

*12-Bromo-2-phenyl-2H,8H-benzo[g]bis(**[1,2,3]**triazolo)[5,1-c:4’,5’-e]**[1,4]**oxazocine* (**4e**). Yield 261 mg (66%), white solid, m. p. 224 °C. IR (ν, cm^−1^): no signals of azido group and triple bond were observed. ^1^H-NMR: δ 5.56 (s, 2H), 7.18 (d, 1H, *J* = 8.7 Hz), 7.45 (tt, 1H, *J* = 7.4, 1.3 Hz), 7.53–7.60 (m, 3H), 7.68 (s, 1H), 7.89 (d, 1H, *J* = 2.5 Hz), 8.19–8.22 (m, 2H). ^13^C-NMR: δ 66.6, 118.2, 118.9, 123.3, 123.4, 128.6, 129.5, 131.9, 132.8, 134.2, 134.6, 137.0, 139.1, 141.5, 154.7. ESI-HRMS (*m/z*): calcd. for (C_17_H_12_BrN_6_O) [M + H^+^] 395.0250, found 395.0252.

*10,12-Di-tert-butyl-2-phenyl-2H,8H-benzo[g]bis(**[1,2,3]**triazolo)[5,1-c:4’,5’-e]**[1,4]**oxazocine* (**4f**). Yield 335 mg (78%), white-yellow solid, m. p. 217 °C. IR (ν, cm^−1^): no signals of azido group and triple bond were observed. ^1^H-NMR: δ 1.37 (s, 9H), 1.47 (s, 9H), 5.39 (s, 2H), 7.44 (t, 1H, *J* = 7.4 Hz), 7.51–7.57 (m, 3H), 7.77 (s, 1H), 7.86 (d, 1H, *J* = 2.5 Hz), 8.23–8.25 (m, 2H). ^13^C-NMR: δ 31.3, 34.7, 35.5, 65.2, 119.0, 121.2, 126.4, 126.9, 128.4, 129.4, 132.6, 134.2, 138.4, 139.1, 140.0, 142.1, 146.8, 153.8. ESI-HRMS (*m/z*): calcd. for (C_25_H_29_N_6_O) [M + H^+^] 429.2397, found 429.2402.

*14-Phenyl-8H,14H-naphtho[1–g]bis(**[1,2,3]**triazolo)[5,1-c:4’,5’-e]**[1,4]**oxazocine* (**4g**). Yield 184 mg (50%), white-yellow solid, m. p. 143 °C. IR (ν, cm^−1^): no signals of azido group and triple bond were observed. ^1^H-NMR: δ 5.69 (s, 2H), 7.43–7.47 (m, 2H), 7.51–7.61 (m, 5H), 7.88 (d, 1H, *J* = 7.8 Hz), 8.01 (d, 1H, *J* = 8.7 Hz), 8.28–8.31 (m, 2H), 8.33 (d, 1H, *J* = 8.5 Hz). ^13^C-NMR: δ 67.4, 111.3, 117.0, 118.9, 119.5, 125.2, 126.1, 128.0, 128.3, 129.4, 131.3, 131.3, 132.7, 134.9, 137.2, 139.4, 143.2, 153.6. ESI-HRMS (*m/z*): calcd. for (C_21_H_15_N_6_O) [M + H^+^] 367.1302, found 367.1309.

*2-(4-Fluorophenyl)-2H,8H-benzo[g]bis(**[1,2,3]**triazolo)[5,1-c:4’,5’-e]**[1,4]**oxazocine* (**4h**). Yield 285 mg (85%), white solid, m. p. 168 °C. IR (ν, cm^−1^): no signals of azido group and triple bond were observed. ^1^H-NMR: δ 5.59 (s, 2H), 7.18–7.24 (m, 2H), 7.27–7.31 (m, 2H), 7.46–7.51 (m, 1H), 7.64 (s, 1H), 7.68–7.70 (m, 1H), 8.15–8.20 (m, 2H). ^13^C-NMR: δ 67.1, 116.1, 116.4, 120.6, 120.7, 121.5, 125.5, 129.9, 131.6, 131.9, 134.7, 135.5, 138.4, 141.7, 155.7, 161.0, 163.4. ESI-HRMS (*m/z*): calcd. for (C_17_H_12_FN_6_O) [M + H^+^] 335.1051, found 335.1050.

*2-(2,4-Dichlorophenyl)-2H,8H-benzo[g]bis(**[1,2,3]**triazolo)[5,1-c:4’,5’-e]**[1,4]**oxazocine* (**4i**). Yield 331 mg (86%), white-yellow solid, m. p. 110 °C. IR (ν, cm^−1^): no signals of azido group and triple bond were observed. ^1^H-NMR: δ 5.61 (s, 2H), 7.27–7.32 (m, 2H), 7.43–7.52 (m, 2H), 7.63–7.64 (m, 2H), 7.69 (dd, 1H, *J* = 7.6, 1.4 Hz), 7.77 (d, 1H, *J* = 8.6 Hz). ^13^C-NMR: δ 67.3, 121.5, 121.5, 125.6, 127.8, 128.2, 130.0, 130.0, 131.0, 131.6, 132.0, 134.7, 136.0, 136.0, 138.9, 142.2, 155.7. ESI-HRMS (*m/z*): calcd. for (C_17_H_11_Cl_2_N_6_O) [M + H^+^] 385.0366, found 385.0373.

*2-(2-Chlorophenyl)-2H,8H-benzo[g]bis(**[1,2,3]**triazolo)[5,1-c:4’,5’-e]**[1,4]**oxazocine* (**4j**). Yield 242 mg (69%), white solid, m. p. 80 °C. IR (ν, cm^−1^): no signals of azido group and triple bond were observed. ^1^H-NMR: δ 5.63 (s, 2H), 7.28–7.33 (m, 2H), 7.45–7.53 (m, 3H), 7.61–7.65 (m, 2H), 7.73 (dd, 1H, *J* = 7.6, 1.6 Hz), 7.80–7.84 (m, 1H). ^13^C-NMR: δ 67.2, 121.5, 121.6, 125.6, 127.5, 127.7, 129.5, 130.1, 130.8, 131.2, 131.6, 131.9, 134.7, 137.5, 138.6, 142.0, 155.7. ESI-HRMS (*m/z*): calcd. for (C_17_H_12_ClN_6_O) [M + H^+^] 351.0756, found 351.0758.

*2-(2-Methoxyphenyl)-2H,8H-benzo[g]bis(**[1,2,3]**triazolo)[5,1-c:4’,5’-e]**[1,4]**oxazocine* (**4k**). Yield 232 mg (67%), white solid, m. p. 65 °C. IR (ν, cm^−1^): no signals of azido group and triple bond were observed. ^1^H-NMR: δ 3.89 (s, 3H), 5.58 (s, 2H), 7.08–7.12 (m, 2H), 7.23–7.29 (m, 2H), 7.43–7.50 (m, 2H), 7.62 (s, 1H), 7.68–7.72 (m, 2H). ^13^C-NMR: δ 56.2, 67.0, 112.6, 120.5, 121.4, 121.8, 125.4, 127.1, 129.0, 130.0, 131.1, 131.5, 131.6, 134.5, 137.9, 141.4, 153.5, 155.6. ESI-HRMS (*m/z*): calcd. for (C_18_H_15_N_6_O_2_) [M + H^+^] 347.1251, found 347.1244.

*2-(4-Methoxyphenyl)-2H,8H-benzo[g]bis(**[1,2,3]**triazolo)[5,1-c:4’,5’-e]**[1,4]**oxazocine* (**4l**). Yield 253 mg (73%), white solid, m. p.65 °C. IR (ν, cm^−1^): no signals of azido group and triple bond were observed. ^1^H-NMR: δ 3.89 (s, 3H), 5.59 (s, 2H), 7.02–7.06 (m, 2H), 7.28–7.31 (m, 2H), 7.46–7.51 (m, 1H), 7.65 (s, 1H), 7.72 (dd, 1H, *J* = 8.0, 1.8 Hz), 8.11–8.15 (m, 2H). ^13^C-NMR: δ 55.6, 67.0, 114.4, 120.4, 121.5, 121.8, 125.5, 130.1, 131.7, 131.7, 133.0, 134.6, 137.8, 141.2, 155.6, 159.6. ESI-HRMS (*m/z*): calcd. for (C_18_H_15_N_6_O_2_) [M + H^+^] 347.1251, found 347.1254.

*(2,4-Dimethylphenyl)-2H,8H-benzo[g]bis(**[1,2,3]**triazolo)[5,1-c:4’,5’-e]**[1,4]**oxazocine* (**4m**). Yield 253 mg (77%), white solid, m. p.173 °C. IR (ν, cm^−1^): no signals of azido group and triple bond were observed. ^1^H-NMR: δ 2.41 (s, 3H), 2.52 (s, 3H), 5.60 (s, 2H), 7.16–7.29 (m, 2H), 7.26–7.31 (m, 2H), 7.46–7.50 (m, 1H), 7.65 (t, 2H, *J* = 4.0 Hz), 7.70 (dd, 1H, *J* = 7.6, 1.6 Hz). ^13^C-NMR: δ 19.0, 21.1, 67.0, 121.5, 121.9, 125.0, 125.5, 127.3, 130.1, 131.6, 131.6, 132.3, 132.4, 134.6, 136.8, 137.6, 139.4, 141.1, 155.7. ESI-HRMS (*m/z*): calcd. for (C_19_H_17_N_6_O) [M + H^+^] 345.1458, found 345.1450.

*(2,6-Dimethylphenyl)-2H,8H-benzo[g]bis(**[1,2,3]**triazolo)[5,1-c:4’,5’-e]**[1,4]**oxazocine* (**4n**). Yield 293 mg (85%), white solid, m. p.175 °C. IR (ν, cm^−1^): no signals of azido group and triple bond were observed. ^1^H-NMR: δ 2.21 (s, 6H), 5.62 (s, 2H), 7.21 (d, 2H, *J* = 7.6 Hz), 7.26–7.37 (m, 3H), 7.46–7.50 (m, 1H), 7.66 (s, 1H), 7.72 (dd, 1H, *J* = 7.7, 1.6 Hz). ^13^C-NMR: δ 17.6, 67.2, 121.6, 121.8, 125.5, 128.4, 130.0, 130.2, 131.6, 131.7, 134.5, 135.7, 137.4, 138.9, 141.1, 155.8. ESI-HRMS (*m/z*): calcd. for (C_19_H_17_N_6_O) [M + H^+^] 345.1458, found 345.1450.

*4-(2H,8H-Benzo[g]bis(**[1,2,3]**triazolo)[5,1-c:4’,5’-e]**[1,4]**oxazocin-2-yl)benzonitrile* (**4o**). Yield 178 mg (52%), white solid, m. p. 198 °C. IR (ν, cm^−1^): no signals of azido group and triple bond were observed. ^1^H-NMR: δ 5.62 (s, 2H), 7.31–7.35 (m, 2H), 7.52–7.56 (m, 1H), 7.65 (s, 1H), 7.70 (dd, 1H, *J* = 8.0, 1.6 Hz), 7.84 (dt, 2H*, J* = 9.0, 2.0 Hz), 8.34 (dt, 2H, *J* = 9.0, 2.0 Hz). ^13^C-NMR: δ 67.4, 111.7, 115.5, 118.0, 119.1, 121.4, 121.6, 125.8, 129.8, 131.6, 132.4, 133.6, 134.9, 139.9, 141.7, 142.9, 155.8. ESI-HRMS (*m/z*): calcd. for (C_18_H_12_N_7_O) [M + H^+^] 342.1098, found 342.1096.

## 4. Conclusions

The synthesis and photophysical properties of a series of new differently substituted 2-phenyl-2*H*,8*H*-benzo[g]bis([1,2,3]triazolo)[5,1-c:4’,5’-e][1,4]oxazocines were investigated. The corresponding dichlorodiazenes containing propargyloxy groups were used as a key starting materials for this aim. Their reaction with sodium azide leads directly to the corresponding 4-azido-1,2,3-triazoles in up to 97% yield. Subsequent thermal cyclization resulted in efficient synthesis of condensed heterocycles having an additional triazole ring in up to 86% yield. The prepared oxazocine derivatives demonstrated interesting photophysical properties and much higher fluorescence quantum yields in comparison to non-cyclized triazole derivatives.

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
