# Peer review of "Synthesis and Spectral Study of a New Family of 2,5-Diaryltriazoles Having Restricted Rotation of the 5-Aryl Substituent"

_molecules, 2020, doi:10.3390/molecules25030480_

Round 1

Reviewer 1 Report

The English in the present manuscript is not of publication quality and requires major improvement

The aim of the research is not clearly mentioned. The biological activity mentioned in the introduction  is not related to the aim of this research.

It is not clear which are the compounds 1a-i and the substituents are not corresponding to Table 1 (Figure 3a has to different substituents).

The subpoints from Results and Discussions are not numbered correctly.

The synthesis schemes should be numbered separately and the obtained compounds should be listed in tables.

The titles of schemes 2, 3, 5 should be reformulated

Meaning of  rt abbreviation

Tabel 2: 4m and 4n are inversed

Author Response

Thank you very much for valuable remarks concerning our manuscript.

The revision was prepared accordingly all your recommendations.

The English in the present manuscript is not of publication quality and requires major improvement

Corrected

The aim of the research is not clearly mentioned. The biological activity mentioned in the introduction  is not related to the aim of this research.

Added

It is not clear which are the compounds 1a-i and the substituents are not corresponding to Table 1 (Figure 3a has to different substituents).

Corrected

The subpoints from Results and Discussions are not numbered correctly.

Corrected

The synthesis schemes should be numbered separately and the obtained compounds should be listed in tables.

Corrected

The titles of schemes 2, 3, 5 should be reformulated

Corrected

Meaning of  rt abbreviation

Added

Tabel 2: 4m and 4n are inversed

Corrected

Reviewer 2 Report

Tsyrenova et al. reported the preparation of a new class of blue-emitting compounds. The synthesis of the molecules has been performed exploiting a method developed by the authors followed by a cyclization. Then the authors studied the photophysical properties of the compounds.

The paper is well written and scientifically clear. Anyway some points that have to be taken care of:

The general structure of compound 1 is missing. Although it is clear from the IUPAC name, I think that the authors can add the structure, for instance, above table 1 Rows 78-79: to me the absorption spectra are not so similar. For instance, if you compare 1a-d it is clear that the electronic properties of the substituents have a strong effect on the abs spectra. The authors should expand this part adding comments Row 86: again the emission spectra are not similar for all the compounds. I can see two trends: one for compounds 1a-d and 1g-h and another for compounds 1e-f, as the authors already highlighted Absorption spectra of compounds 4: although it is not always easy rationalized the spectra I only partially agree with the authors regarding the compounds 4i-o. Compounds 4l and 4o have a bathochromic shift, on the other hand compounds 4i-k and 4m-n have a hypsochromic shift. There is not a correlation with the electronic effects of the substituents. In fact 4o has an EWG, while 4l has an EDG. The hypsochromic group has both substituents with moderate EWG (halogen) and moderate EDG (methyl). Abs spectrum of 4g belongs to the “bathochromic group”, while the spectrum of 4h belongs to the group with Ph in position 2. Are the author sure that they did not switched the colour accidentally? NMR: the details of the instrument missing The experimental part regarding the measurement of the quantum yield is missing

Author Response

Thank you very much for valuable remarks concerning our manuscript.

The revision was prepared accordingly all your recommendations.

The general structure of compound 1 is missing. Although it is clear from the IUPAC name, I think that the authors can add the structure, for instance, above table 1

added

Rows 78-79: to me the absorption spectra are not so similar. For instance, if you compare 1a-d it is clear that the electronic properties of the substituents have a strong effect on the abs spectra. The authors should expand this part adding comments Row 86: again the emission spectra are not similar for all the compounds. I can see two trends: one for compounds 1a-d and 1g-h and another for compounds 1e-f, as the authors already highlighted Absorption spectra of compounds 4: although it is not always easy rationalized the spectra I only partially agree with the authors regarding the compounds 4i-o. Compounds 4l and 4o have a bathochromic shift, on the other hand compounds 4i-k and 4m-n have a hypsochromic shift. There is not a correlation with the electronic effects of the substituents. In fact 4o has an EWG, while 4l has an EDG. The hypsochromic group has both substituents with moderate EWG (halogen) and moderate EDG (methyl). Abs spectrum of 4g belongs to the “bathochromic group”, while the spectrum of 4h belongs to the group with Ph in position 2. Are the author sure that they did not switched the colour accidentally?

corrected

NMR: the details of the instrument missing The experimental part regarding the measurement of the quantum yield is missing 

corrected

Round 2

Reviewer 1 Report

The article can be published.